# Associated factors, health-related quality of life, and reported costs of chronic otitis media in adults at two otologic referral centers in a middle-income country

**Lucia C. Pérez-Herrera**[1,2], **Daniel Peñaranda**[3], **Sergio Moreno-López**[1,2], **Ana M. Otoya-Tono**[3], **Lorena Gutiérrez-Velasco**[3], **Juan Manuel García**[2,3,4], **Augusto Peñaranda**[2,4]*

**1** Otolaryngology and Allergology Research Groups, Unidad Médico Quirúrgica de Otorrinolaringología (UNIMEQ-ORL), Bogotá, Colombia, **2** Faculty of Medicine, Universidad de Los Andes, Bogotá, Colombia, **3** Otolaryngology Department, Fundación Universitaria de Ciencias de la Salud, Bogotá, Colombia, **4** Otolaryngology Section, Hospital Universitario Fundación Santa Fe de Bogotá, Bogotá, Colombia

* augpenar@gmail.com

## Abstract

### Background

Despite the high prevalence of chronic otitis media (COM) in low to middle-income countries, there are few studies regarding its associated factors, health-related quality of life, and treatment costs. This study aimed to identify associated factors of COM, assess its impact on the quality of life as well as estimate the patients' reported costs of COM treatment in Colombia.

### Methods

Cross-sectional study. Two otology-referral centers in Bogotá (Colombia) were included. Questionnaires focusing on sociodemographic and clinical associated factors, quality of life, and patients' reported costs were administered to 200 adults with COM diagnosis and 144 control adults. Otoscopic evaluation and audiometric data were collected.

### Results

The mean age was 42.2 years (SD: 14.44). The median length of COM was 26.13 years (SD: 17.06), and 79.5% of the COM patients reported otorrhea during childhood (P-value: 0.01). The most frequently reported allergic disease among our study population was allergic rhinitis (26.5%). COM was less frequent in patients with a medium-high socioeconomic status (PR: 0.54; 95% CI: 0.39–0.72), and more frequent in patients who reported increased ear discharge due to upper respiratory tract infections (PR: 1.69; 95% CI: 1.68–1.70). The global score of the "Chronic Suppurative Otitis Media Questionnaire-12" showed a difference of 9 points between patients with active and inactive COM (P < 0.001). Patients spent between 12.07% to 60.37% of their household income on expenses related to COM.

**Data Availability Statement:** All relevant data are within the manuscript.

**Funding:** This project was funded by the Unidad Medico Quirúrgica de Otorrinolaringología-UNIMEQ-ORL from Bogotá, Colombia. URL: https://www.unimeqorl.co/. The funders had no role in study design, data collection and analysis, decision to publish, or preparation of the manuscript.

**Competing interests:** The authors have declared that no competing interests exist.

**Abbreviations:** AOM, Acute Otitis Media; COM, Chronic Otitis Media; COMQ-12, Chronic Otitis Media Questionnaire 12; COP, Colombian Pesos; dB, decibels; ENT, Ear, Nose and Throat specialist; HRQoL, Health-Related Quality of Life; OR, Odds Ratio; PR, Prevalence Ratio; SD, Standard Deviation; URTI, Upper respiratory Tract Infections; USD, United States Dollar; VAS, Visual Analogue Scale.

## Conclusions

Associated factors found in this study are consistent with previous reports. COM has a significant financial impact and affects patients' quality of life. Worldwide research addressing these issues in poor-resource countries is scarce, further studies are needed.

## Introduction

Chronic otitis media (COM) is a persistent inflammatory condition involving the middle ear and mastoid cavity [1]. COM is also often associated with tympanic membrane perforation [1, 2]. Worldwide COM affects about 2% of the population [3]. Previous studies describe a global downward trend in incidences of COM and complications due to current advances in vaccination and treatment of infectious diseases [3, 4]. However, COM's prevalence remains high in low and middle-income countries, probably due to inadequacies of public health policies; poor hygienic conditions; limited access to health services; and the dearth of ear, nose, and throat (ENT) specialists [2, 5].

In Latin America, there is a shortage of studies available regarding the prevalence of COM and its associated factors [6]. In Colombia, the Integrated Information System for Social Protection (SISPRO) reported a total of 20.777 cases of COM in 2017, and an estimated prevalence of 0.131% [7]. However, these statistics could be underestimated as a result of the high rates of under-reporting, outsourcing in the healthcare system, and the access barriers to healthcare services [8, 9]. Likewise, one of these access barriers to healthcare services is the inequality in the distribution of ENT specialists in Colombia: the mean of ENT's distribution per million people is only 11.8 (95% CI: 10.9 to 12.8) [10].

Indeed, evidence regarding sociodemographic and clinical associated factors associated with COM is scarce and varies widely in methodological quality and sampling methods [2]. Besides, most of these studies are focused on pediatric populations [2, 11]. Considering that recurrent acute otitis media (AOM) may predispose an individual to COM, most of the COM studies extrapolated the associated factors of AOM to perform research on COM [1]. Nevertheless, current evidence reinforces that some of these associations are not reliable, and environmental associated factors are not entirely established yet [11]. A meta-analysis identified statistically significant risk factors for COM including snoring (OR, 1.96; 95% CI: 1.78–2.16; P<0.001), passive smoke exposure (OR, 1.39; 95% CI: 1.02–1.89 P = 0.04), and allergies or atopy (OR, 1.36; 95% CI: 1.13–1.64;P = 0.001) [12].

Further risk factors were identified as well, such as: belonging to a low socio-economic status (OR, 3.82; 95%CI, 1.11–13.15; P = 0.03), and upper respiratory tract infections (URTI) (OR, 6.59; 95%CI, 3.13–13.89; P<0.001) [12]. Additional risk factors significantly related to COM include difficult access to healthcare services, a family history (parents) of COM, household overcrowding [13], breastfeeding [14], and genetic factors [15, 16]. Similarly, undernutrition and poor hygienic conditions are also associated with COM, and recent studies suggest that the improvement of those circumstances could lead to a reduction of almost 50% of COM cases [17].

Furthermore, COM symptoms (recurrent smelly discharge, hearing loss, tinnitus, and balance abnormalities) are related to a significant quality of life deterioration due to social communication difficulties and lower work performance [2, 3, 18]. Therefore, the health-related quality of life (HRQoL) measurement in COM patients is considered an important aspect of clinical research to guide therapeutic decision making [19]. "Chronic Suppurative Otitis Media Questionnaire-12" (COMQ-12) is an internationally recognized HRQoL scale for

COM and has been validated in several languages and for differing socio-cultural backgrounds [3, 20]. Although COMQ-12 having been recently validated in the Spanish language [21], there is still no reliable information regarding the impact of COM on patients' quality of life in Hispanic countries.

The economic burden of COM has gone through a significant reduction after the introduction of pneumococcal vaccines in developed countries [22]. Nevertheless, COM's economic burden remains high among adults from low to middle-income countries mostly due to the high direct costs related to this disease [2]. Direct costs of COM include the use of resources for the treatment and care procedures due to the disease such as medications, ENT consults, surgical and non-surgical procedures, hospital stays, diagnostic exams, and supplies [1, 2, 22]. Direct costs related to chronic diseases also include "out-of-pocket expenditures" or non-medical costs derived from the disease which are covered directly by the patients' household income (e.g. transportation to the medical care center expenses) [2, 22, 23]. Additionally, direct costs include the measuring of intangible costs, such as quality of life deterioration, social communication difficulties, anxiety, and reduced work performance [2, 3, 18]. There is no reliable information about the direct costs of COM in Latin America. This information is essential to guide public health interventions aimed to reduce the financial burden of COM and to provide patients with better care.

Considering the gaps in the scientific literature mentioned above, there is no reliable information regarding the prevalence, associated factors of COM, COM's impact on the quality of life, and its patients' reported treatment costs and economic burden in Latin America. A better knowledge of this information across ages and geographical regions is essential to assess the need for interventions in order to reduce COM's social, health, and economic burden [2]. Therefore, the aim of this research was to identify the associated clinical and sociodemographic factors of COM, assess the health-related quality of life of COM patients, and estimate the costs reported by COM patients who visited the Hospital Universitario Fundación Santa Fe de Bogotá and Hospital de San José.

## Materials and methods

### Study design

An observational, cross-sectional, analytical study was performed. The study enrolled patients who visited the Division of Otorhinolaryngology from the Hospital Universitario Fundación Santa Fe de Bogotá or the Hospital de San José between August 2018 and August 2019. Otoscopic evaluations were conducted and questionnaires regarding sociodemographic information, treatment expenses, and quality of life were administered to adult patients. Both institutions are highly complex academic hospitals providing access to all medical specialties and are referral centers for ENT and otology patients from all over the country. These institutions are both private teaching hospitals located in Bogotá, the capital city of Colombia, a low/middle-income country in Latin America. Patients from all over the country visit these institutions seeking for otology health-care.

For this study, we based on the definition of COM as a chronic inflammation of the middle ear and mastoid which persists over 6 weeks to 3 months despite medical treatment [1]. Moreover, COM diagnosis included a broad spectrum of clinical features such as hearing impairment symptoms (e.g. otalgia, tinnitus, persistent blockage of fullness of the ear, hearing loss, chronic ear drainage, balance problems), otoscopic findings to address disease activity, and pure-tone audiometric findings [24].

Ethics committee approval was received for this study from the ethics committee of the Hospital Universitario Fundación Santa Fe de Bogotá (CCEI-9549-2018), and the Hospital de

San José (act number 500, DI-I-0632-18) according to the Helsinki Declaration. Written informed consent was obtained from all participants included in the study.

## Participants and data

A minimum sample size of 278 patients was estimated based on the previously mentioned meta-analysis and including a 15% loss rate [12]. This size provided an odds ratio in the population of low strata of 3.82 [12], a significance level of 5%, a power of 90% [25], and an "exposed/non-exposed ratio" of 1. Regarding the sample selection method, we conducted a non-probability consecutive sampling of the subjects who met the following inclusion criteria: Two hundred patients aged 18 years-old or above with COM diagnosis who attended the Hospital Universitario Fundación Santa Fe de Bogotá or Hospital de San José, agreed to participate and signed the informed consent of the study. Furthermore, one hundred forty-four patients aged 18 years-old or above with normal otoscopic and audiometric findings and without middle ear diseases were also included. These control adults who consulted the otology department had a diagnosis of benign paroxysmal positional vertigo or mandibular joint dysfunction. Overall, the 344 patients were enrolled in the study.

Once the purpose of the study was explained, written informed consent was obtained from all individual participants included in the study. Next, they were examined to confirm the clinical and otoscopic diagnosis of COM and then completed the sociodemographic, clinical associated factors, and quality of life (COMQ-12) questionnaires. The questionnaires were administered and collected by trained researchers with wide experience on the use of these tools. Two otologists, an otolaryngology resident and a general practitioner from both medical institutions included in the study performed the otoscopic examination. All the audiometric testing results were brought by the patients and then recorded in the database. Air and bone conduction thresholds in decibels (dB) values for conventional pure tone frequencies (0.25–4 kHz) were gathered.

Exclusion criteria were as follows: Patients who had other diseases affecting the middle ear (Otosclerosis, tumors or congenital abnormalities of the middle ear), history of head injury, meningitis, previous ear surgery, severe comorbidities (e.g. cancer, HIV), psychiatric disorders, cognitive deficits, or any medical condition that would limit their ability to participate were not included. Patients who refused to sign the informed consent were not included in the study either.

## Sociodemographic and clinical questionnaire

The demographic and clinical questionnaire utilized was a survey that sought information about previously described risk factors in adults with COM. The information included in the questionnaire was: biodata such as sex, age, and body mass index (BMI), number of people living in the household, socioeconomic status, educational level, presence of allergic diseases, childbirth, breastfeeding information, household income, COM's related costs, number of ENT consultations, history of ear discharge during childhood, disease duration, a relationship between increased ear discharge (otorrhea episodes) and URTI, smoking, and presence of COM in other siblings.

Both institutions included in this study treat population affiliated to private Health Promoting Entities which provide health insurance packages to all socioeconomic-status populations. The socioeconomic status was classified as low-income levels (strata I and II), middle-income levels (strata III and IV), and high-income levels (strata V and VI) based on the strata classification of the National Administrative Department of Statistics (DANE) from Colombia. This department uses income data, property base information, and residential characteristics of

people's homes to establish this stratification [26]. Conversely, household income is more indicative of an economic standard of living and is widely used to determine the economic health of an area [27]. For the purpose of this study, we refer to the definition of household income as the incomes of all people occupying the same housing unit, regardless of their familiar relationship [27]. Thus, a single person occupying a dwelling by himself was also considered a household.

Household income and COM's related cost information were reported based on the official minimum wage rates from Colombia, which is $252.41 United States Dollar (USD) or $828,166 Colombian Pesos (COP) per month. COM's related costs were defined as the amount of money directly paid by the patients for their topical treatment, transportation to medical care, or any expenses spent on COM's medical treatment. Likewise, the basic data definition included allergy diseases like the presence or history of allergic rhinitis or asthma or atopic dermatitis, and URTI such as the presence of cough or rhinorrhea or nasal stuffiness or sore throat or flu-related symptoms [12]. BMI categories were defined following the WHO recommendations [28].

## Otoscopic findings

Two otologists, an otolaryngology resident, and a general practitioner performed the otoscopic examination using a Welch-Allyn Otoscope. The otoscopic findings were classified into 5 main categories attaching to Fonseca *et al* recommendations [24]. To ease the statistical analysis, we sorted these categories into two main clusters according to the disease activity as follows:

- Active COM: Active squamous epithelium (cholesteatoma), and perforated eardrum with discharge.

- Inactive COM: Inactive squamous epithelium (retraction, atelectasis, epidermolysis), dry perforated eardrum, and healed eardrum (neo-tympanum, intact tympanic membrane, tympanosclerosis).

## COMQ-12 questionnaire

The original version of COMQ-12 in English was developed and validated by Phillips et al [3]. This questionnaire has gained international recognition due to its strong psychometric properties and its usefulness for assessing quality of life in patients with COM [18, 20]. The Spanish version of COMQ-12 that was used in this study has been previously adapted and validated in the Spanish language [21]. The Spanish version of the COMQ-12 questionnaire includes 12 self-assessment questions grouped in four categories: questions 1 through 7 are related to the severity of symptoms, questions 8 and 9 inquire about the impact of the disease on work and lifestyle, questions 10 and 11 measure the impact of COM on health services, and question 12 is a visual analog scale (VAS) ranging from 0 to 5 that quantifies the global quality of life. Each item is scored from 0 (no quality of life disturbance) to 5 (severe quality of life disturbance) according to the level of discomfort indicated by the patient.

## Statistical analysis

Statistical analysis was performed using Stata 16MP software. Relative and absolute frequencies for qualitative variables were calculated as well as a central trend and dispersion measures. A Fisher or Chi-squared test was conducted to determine the associations between qualitative variables. A t-Student or Mann Whitney test was also performed to evaluate the associations

between qualitative and quantitative variables according to data distribution. Additionally, a robust logistic regression analysis of the clusters and a transformation of the odds ratios (OR) into prevalence ratios (PR) were also carried out to find the adjusted effect of the possible variables associated with the presence of COM. Since each COM patient's ear had an independent audiometric result, the actual sample size to carry out the robust logistics regression analysis was the available data of 400 affected ears. Bivariate and multivariate analysis of the study variables was used to evaluate the correlations (via PR) between the associated factors and the presence of COM. Model assumptions were validated through a linearity test, the Hosmer-Lemeshow test, an estimation of deviance residuals and leverage values, and a comparison between the crude and the adjusted models. Hypothesis testing to determine the level of statistical significance was done using a 95% confidence interval and a P value less than 0.05 (P<0.05).

In the group of COM patients, the audiometric test values for conventional frequencies (0.25–4 kHz) were used to calculate air-bone gap thresholds, that is the difference between bone conduction and air conduction thresholds. These calculations were segregated by affected ear and were classified according to the activity of the disease (active or inactive COM). Likewise, a Mann-Whitney test was performed, and with the obtained scores of the COMQ-12, was used to estimate the difference between obtained scores in the Active and inactive COM groups. A descriptive analysis of household-income and treatment costs reported by COM patients was carried out. Income and related cost information were reported based on the official minimum wage rates from Colombia, which is $252.41 USD ($828,166 COP) per month. We converted these values to the US dollar using the 2009 annual average exchange rate ($1 USD = $3,281.09 COP) [29].

## Results

A total of 344 patients who met the eligibility criteria were included in the study, 200 with a COM diagnosis and 144 as controls without middle ear disease. Tables 1 and 2 describe the demographic and clinical characteristics of the patients. Statistical differences were found for socioeconomic variables such as "socioeconomic status", "educational level", "breastfeeding length", "ear discharge during childhood", and "increased ear discharge due to URTI". The median number of people living in the household was 3 people, no differences were found between the study groups (P = 0.06).

Bivariate and multivariate analysis of the sociodemographic and clinical variables related to COM are shown in Table 3. Belonging to medium or high socioeconomic status was related to a lower frequency of COM (PR: 0.54; CI 95%:0.39–0.72), as well as female sex (PR: 0.66; CI 95%:0.49–0.84). In contrast, a higher frequency of COM was found in the patients that reported a that ear discharge episodes increased when due to URTI (PR: 1.69; CI 95%:1.68–1.7). No collinearity problems were found trough the linearity and the goodness-of-fit tests. Both tests showed good model specification. Likewise, no extreme or influential values were found for the residuals and leverage values.

### Otoscopic findings

Table 4 describes the otoscopic findings from the COM patients. As previously mentioned, the otoscopic findings were classified into two groups according to the disease activity. It was found that 38% of the population with COM had active disease and the remaining population had inactive disease (62%).

**Table 1. Baseline demographic characteristics of the participants.**

| Characteristics | COM patients N = 200 | | Control group N = 144 | | Total N = 344 | | P- Value[a] |
|---|---|---|---|---|---|---|---|
| | N | % | N | % | N | % | |
| Sex, f/m | 105/95 | 52.5/47.5 | 86/58 | 59.7/40.3 | 191/153 | 55.5/44.5 | 0.184 |
| Age (Years) [b] | 41.8 (15.3) | 40.8 (18–85.5) | 42.6 (13.2) | 41.4 (18.1–82.3) | 42.2 (14.4) | 40.8 (18–85.5) | 0.5 |
| Socioeconomic Status | | | | | | | 0.01 |
| • Low-income levels | 139 | 69.5 | 66 | 45.8 | 205 | 59.6 | |
| • Medium-income levels | 58 | 29 | 77 | 53.5 | 135 | 39.3 | |
| • High-income levels | 3 | 1.5 | 1 | 0.7 | 4 | 1.2 | |
| Educational level | | | | | | | 0.01 |
| • Primary education | 33 | 16.5 | 17 | 11.8 | 50 | 14.5 | |
| • Secondary education | 104 | 52 | 54 | 37.5 | 158 | 45.9 | |
| • Higher education | 63 | 31.5 | 73 | 50.7 | 136 | 39.6 | |
| Total number of people living in the household [b] | 3 (1–7) | - | 3 (1–8) | - | 3 (1–8) | - | 0.06 |
| BMI (kg/m$^2$) [b] | 25.2 (4.19) | 24.7 (15.8–42.7) | 24.7 (3.46) | 24.5 (16.5–35.5) | 25 (3.9) | 24.6 (15.8–42.7) | 0.42 |
| BMI category | | | | | | | 0.77 |
| • Underweight | 2 | 1 | 2 | 1.4 | 4 | 1.2 | |
| • Normal weight | 105 | 52.5 | 81 | 56.3 | 186 | 54.1 | |
| • Overweight | 72 | 36 | 50 | 34.7 | 122 | 35.5 | |
| • Obese | 21 | 10.5 | 11 | 7.6 | 32 | 9.3 | |

[a.] P values obtained through the Wilcoxon Rank test or Fisher test considering the characteristics of each variable.

[b.] Values are expressed in Mean (SD) and Median (Range).

### Disease activity and audiometric test results

The audiometric values for conventional pure tone frequencies (0.25–4 kHz) segregated by affected ear according to the disease activity (active or inactive COM) are shown in Figs 1 and 2. Mean and median of air and bone conduction for each frequency were reported considering that there is a lack of statistical consensus on which central tendency measure should be used to describe the hearing range. Conductive hearing loss was found in both groups, and a 30 dB air-bone gap was found in patients with active COM. The air-bone gap for low frequencies (0.5–1 kHz) in patients with active COM was slightly higher compared to the population with inactive disease. However, this difference between both groups does not exceed 5 dB.

### Patients' reported COM costs

Up to 41% of the population with COM reported an estimated household income between $282.00 USD ($925,251 COP) and $563.90 USD ($1,850,175 COP) per month. The reported expenditures for COM treatment per month including transportation to medical care centers oscillated between 6.03% and 24.15% of the patients' household monthly income, equivalent to $252.41 USD ($828,116 COP). Moreover, the expenses spent on the disease over 6 months increased to a range between 12.07% to 60.37% of their household income during that time. Patients reported a median of 5 consultations with an ENT specialist in a time interval of 6 months. All these results are shown in Table 5.

### Quality of life of COM patients

We sought to determine the possible correlation between disease activity and health-related quality of life using the COMQ-12 scale and its domains. Comparison between the domains of the scale according to disease activity revealed that there was no difference for the "general

**Table 2. Clinical characteristics of the participants.**

| Characteristics | COM patients N = 200 | | Control group N = 144 | | Total N = 344 | | P- Value[a] |
|---|---|---|---|---|---|---|---|
| | N | % | N | % | N | % | |
| Childbirth | | | | | | | 0.92 |
| • Cesarean section | 24 | 12 | 16 | 11.1 | 40 | 11.6 | |
| • Vaginal Delivery | 176 | 88 | 128 | 88.9 | 304 | 88.4 | |
| Breastfeeding in infancy | | | | | | | 0.66 |
| • Yes | 185 | 92.5 | 135 | 93.8 | 320 | 93.0 | |
| • No | 15 | 7.5 | 9 | 6.3 | 24 | 7.0 | |
| Breastfeeding length | | | | | | | 0.01 |
| • Less than 6 months | 38 | 19 | 12 | 8.3 | 50 | 14.5 | |
| • More than 6 months | 147 | 73.5 | 123 | 85.4 | 270 | 78.5 | |
| • Without breastfeeding | 15 | 7.5 | 9 | 6.3 | 24 | 7.0 | |
| Ear discharge during childhood | | | | | | | 0.01 |
| • Yes | 159 | 79.5 | 0 | 0.0 | 159 | 46.2 | |
| Increased ear discharge due to URTI [b] | | | | | | | |
| • Yes | 135 | 67.5 | 5 | 3.5 | 140 | 40.7 | 0.01 |
| Allergic Disease | | | | | | | |
| • Allergic Rhinitis | 49 | 24.5 | 42 | 29.2 | 91 | 26.5 | 0.33 |
| • Asthma | 13 | 6.5 | 8 | 5.6 | 21 | 6.1 | 0.71 |
| • Atopic Dermatitis | 8 | 4 | 5 | 3.5 | 13 | 3.8 | 0.80 |
| Smoking | | | | | | | |
| • Yes | 26 | 13 | 20 | 13.9 | 46 | 13.4 | 0.94 |
| Family history of COM | | | | | | | |
| • Paternal history | 9 | 4.5 | 13 | 9.0 | 22 | 6.4 | 0.12 |
| • Maternal history | 8 | 4 | 4 | 2.8 | 12 | 3.5 | 0.77 |
| Disease duration (Years) [c] | 26.13 (17.06) | 23 (1–63) | 0 | 0.0 | 26.13 (17.06) | 23 (1–63) | NA |
| Active COM | | | | | | | |
| • Yes | 76 | 38 | – | – | 76 | 38.0 | NA |
| • No | 124 | 62 | – | – | 124 | 62.0 | NA |

[a.] P values obtained through the Wilcoxon Rank test or Fisher test considering the characteristics of each variable.

[b.] URTI included the presence of cough or rhinorrhea or nasal stuffiness or flu-related symptoms.

[c.] Values are expressed in Mean (SD) and Median (Range).

appearance" domain (P = 0.078); thus, the hypothesis of equality of medians was not rejected. A Mann Whitney test was carried out to evaluate the remaining domains of the scale, which suggested possible differences between the compared groups (P<0.05). The median differences between these domains were 6.5 points for the "severity of disease" domain, and 1 point for the "impact on lifestyle and work" domain. Additionally, the global score also presented differences between the study groups with a difference of 9 points (P<0.001). These results are described in Table 6.

## Discussion

This study aimed to identify associated factors of COM, assess the impact of this disease on the quality of life as well as estimate the patients' reported costs of treating COM. Our results contribute to the understanding of the modifiable associated factors of COM in a representative sample of adult patients from low to middle-income countries. COM is the most frequent

Table 3. Associated factors related to COM.

| Variable | Bivariate analysis | | Multivariate analysis [b] | |
|---|---|---|---|---|
| | PR [a] | CI 95% | PR [a] | IC 95% |
| Age | 1.00 | (0.99–1.00) | 1.00 | (0.99–1.01) |
| Ear | | | | |
| • Left ear | 1.00 | (0.87–1.12) | 1.00 | (0.82–1.17) |
| Sex | | | | |
| • Female | 0.89 | (0.76–1.00) | 0.66 | (0.49–0.84) |
| Socioeconomic status | | | | |
| • Medium (3–4) and high (5–6) income levels | 0.66 | (0.54–0.77) | 0.54 | (0.39–0.72) |
| History of Allergic rhinitis | | | | |
| • Yes | 0.90 | (0.76–1.04) | 0.75 | (0.52–1.00) |
| Asthma history | | | | |
| • Yes | 1.07 | (0.80–1.30) | 0.89 | (0.46–1.30) |
| Atopic dermatitis history | | | | |
| • Yes | 1.06 | (0.72–1.35) | 1.07 | (0.57–1.45) |
| Breastfeeding during childhood | | | | |
| • Yes | 0.95 | (0.70–1.18) | 1.00 | (0.61–1.33) |
| Smoking history | | | | |
| • Yes | 0.97 | (0.78–1.14) | 0.93 | (0.67–1.17) |
| COM in father | | | | |
| • Yes | 0.69 | (0.46–0.95) | 0.82 | (0.45–1.19) |
| COM in mother | | | | |
| • Yes | 1.15 | (0.79–1.43) | 0.97 | (0.48–1.38) |
| Increased ear discharge due to URTI | | | | |
| • Yes | 3.02 | (2.92–3.08) | 1.69 | – 1.70) |

[a.] PR: Prevalence Ratio

[b.] Log-likelihood Model: -263.512; AIC:553.024; BIC: 611.67; n = 672

cause of persistent mild to moderate hearing loss among young populations in low to middle-income countries [1]. This scenario could explain the statistical differences between the COM patients and the control group for the "educational level" variable (P = 0.01). The persistent and significant hearing loss derived from COM has been linked to poor scholastic performance and social communication difficulties [1–3, 18]. Moreover, the lack of access to hearing aids worsens these hearing disabilities leading to learning difficulties and lower academic attainments [1].

Regarding the breastfeeding length differences found among the groups (P = 0.01), several studies suggest that lack of breastfeeding and inadequate patterns of breastfeeding are associated with increased risk of COM [14]. However, this association was not found in this study probably due to the high frequency of breastfeeding in the study population (93%) and the recall bias related to this variable. Furthermore, 79.5% of COM patients reported a history of childhood ear drainage (P = 0.01). Previous studies suggest that a history of ear drainage in childhood is considered a major risk factor for the development of OMC [30, 31]. However, none of the patients in the control group reported this medical history; therefore, the role of this variable as an associated factor for the development of COM could not be established in this study. Finally, it is remarkable that the average disease duration reported by the patients reaches 26.13 years (SD: 17.03). This scenario displays the financial barriers impact on access

**Table 4. COM patient otoscopic findings.**

| Disease activity | Otoscopic findings | Right ear | | Left ear | |
|---|---|---|---|---|---|
| | | N | % | N | % |
| Active COM | Active squamous epithelium (Cholesteatoma) | 15 | 7.5 | 19 | 9.5 |
| | Perforated eardrum with discharge | 56 | 28 | 51 | 25.5 |
| Inactive COM | Inactive squamous epithelium (retraction, atelectasis, epidermolysis) | 16 | 8 | 14 | 7 |
| | Dry perforated eardrum | 39 | 19.5 | 34 | 17 |
| | Healed eardrum (Neo-tympanum, intact tympanic membrane, tympanosclerosis) | 22 | 11 | 20 | 10 |

to healthcare, the inaccessibility of ENTs in low to medium-resource countries, as well as the therapeutic challenges, the quality of life deterioration, and the economic impact of COM on this group of patients [2, 5].

The most frequently reported allergic disease among our study population was allergic rhinitis. The frequency of allergic rhinitis in this population ranged between 24.5% and 29.3%, which is similar to previous studies in Colombian populations that reported a frequency between 29.5% and 33.9% [32]. Likewise, previous international studies describe a frequency of allergic rhinitis ranging from 24% and 89% in the general population [12]. Indeed, some authors state that respiratory allergies such as allergic rhinitis contribute to the onset of COM. Thus, the results in our study population are consistent with previous findings.

Regarding the factors associated with COM, we found a statistically significant association between the socioeconomic status and the development of COM: a lower frequency of COM was found in patients belonging to a medium to high socioeconomic status (PR: 0.54; 95% CI: 0.39–0.72). Similarly, it has been reported that belonging to a low socioeconomic status is an important risk factor for the development of OMC compared to belonging to a middle or high status [12, 15, 33, 34]. This association could be explained by the poor nutrition, hygiene conditions, and the healthcare access barriers that mainly affect populations of the low socioeconomic status [33]. Overall, this population is more predisposed to COM as well as experiences an increased rate of COM complications. In addition, overcrowding and poor access to education about healthy habits cause higher recurrence rates of URTI, which is also a reported risk

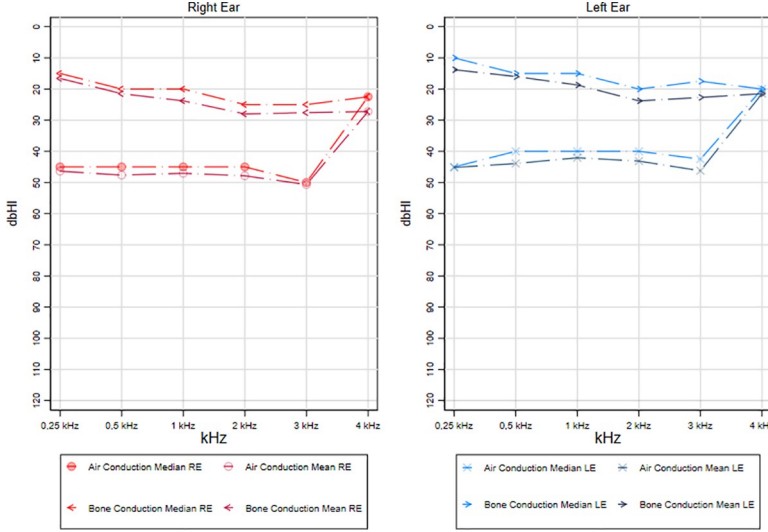

**Fig 1. Audiometric test results on the active COM group.**

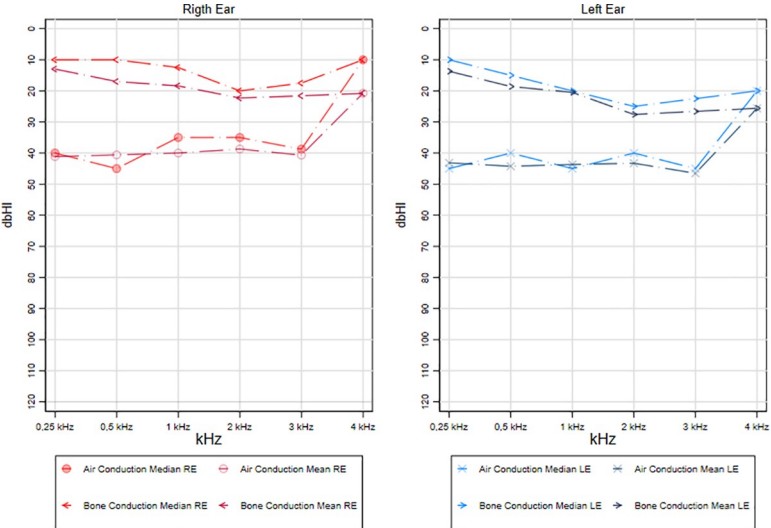

**Fig 2. Audiometric test results on the inactive COM group.**

factor for the development of COM [34]. Considering previous studies suggesting that improvement of these conditions could reduce the frequency of COM [17], public health policies should be designed focusing on improving these conditions.

Similarly, a higher frequency of COM was found in patients who reported a relationship between URTI and increased ear discharge or otorrhea symptoms (PR: 1.69; 95% CI: 1.68–1.7). This association is clinically and statistically significant and has been previously reported in the literature [12, 30, 31, 35, 36]. As an illustration, a COM risk factors meta-analysis performed by Zhang *et al* showed pooled data that suggested that the presence of URTI significantly increased the risk of COM (OR, 6.59; 95% CI, 3.13–13.89; P<0.001) [12]. Likewise, in a cohort of 465 children who were followed over 2 years to establish risk factors associated with COM, Koch *et al* identified the presence of URTI as an significant risk factor for COM (HR: 1.19, 95% CI: 1.03–1.37) [31]. Additionally, on 300 microbiological samples from patients with different types of otitis media (AOM, COM and otitis media with effusion), Hassooni *et al* reported that an association between the microbiological findings of the different types of otitis

**Table 5. Household income and costs related to COM.**

| Variables | N | % |
|---|---|---|
| Household monthly income (USD) [a, b, c] | | |
| • 282 to 563.90 | 82 | 41 |
| • 564 to 845.90 | 77 | 38.5 |
| • More than 846 | 41 | 20.5 |
| COM related costs during the last month (USD) [c] | 30.48 | (15.24–60.96) |
| COM related costs during the last 6 months (USD) [c] | 76.19 | (30.48–152.39) |
| Percentage of the household income spent on COM on the last month | 12.08 | (6.04–24.15) |
| Percentage of the household income spent on COM during the last six months | 30.19 | (12.08–60.38) |
| Number of ENT or otolaryngology consultations in the previous 6 months [a] | 5 | (3–8) |

[a.] Values are expressed in Median (Range) and IQR (Interquartile range: interval from percentile 25 to 75).

[b.] USD 282 is the equivalent value for the official minimum monthly wage rate from Colombia in 2019.

[c.] Mean TRM 2019 $3281.09 in Colombian pesos.

**Table 6. Domain values of the COMQ-12 scale based on disease activity.**

| Domain | Active COM | | Inactive COM | | |
|---|---|---|---|---|---|
| | Median | IQR[(a)] | Median | IQR[(a)] | p-value [(b)] |
| Symptom severity [a] | 27 | (20.5–31.5) | 20.5 | (14.5–26.5) | <0.001 |
| Impact on lifestyle and work [b] | 5 | (3–8) | 4 | (1–6.5) | 0.015 |
| Impact on health services [c] | 8 | (7–8.5) | 8 | (7–8.5) | 0.996 |
| VAS [c] | 4 | (3–5) | 4 | (3–5) | 0.078 |
| Total [a] | 44 | (35–51.5) | 35 | (26.5–45) | <0.001 |

[a.] IQR: Interquartile range or interval formed from the 25th to the 75th percentile

[b.] Based on a Mann Whitney rank test

media and URTI was found in up to 42% of the samples [35]. Thus, our findings regarding this association have biological plausibility and are supported in previous studies.

The frequency of otoscopic findings described in COM patients with active and inactive disease is similar to previous studies [24, 37]. Thus, in the group of patients with active COM, a higher frequency of tympanic perforations with otorrhea was found while in patients with inactive COM, the most common otoscopic finding was dry tympanic perforations. Although there are some differences within the reported literature regarding the frequency of the otoscopic findings described [24], the higher frequency of dry perforations and perforations with ear discharge or otorrhea is consistent within these reports. Regarding the frequency of COM with cholesteatoma in our study sample, it varies from 7.5% to 9.5% and is also similar to that described in previous literature [24, 37].

Likewise, the audiometric testing results classified by the disease activity are consistent with previous literature that suggests a mild to moderate conductive hearing loss in patients with COM [37–39]. Moreover, previous reports describe an air-bone gap secondary to conductive hearing loss as a common finding in COM patients [37]. This conductive hearing loss caused by COM could result in a 30 dB air-bone gap [37], which was confirmed in our audiometric testing analysis in patients with active COM. No major differences in the air-bone gap were found between patients with active and inactive disease. Despite many studies have suggested an association between COM and sensorineural hearing loss [40], there is still no consensus about this association. The presence of active disease was not associated with a significantly increased risk of sensorineural hearing loss. However, we should mention that the audiometric tests were carried out in external audiology clinics, and afterwards brought to the otology consult at our institutions. Audiometric results from different clinics may show significant variations due to sound pressure calibration between different audiometers (between test variation) [41]. Indeed, previous studies report that a patient could expect a variation of ±11 dB in the results at some frequencies if tested in different clinics [41].

Considering the information related to patients' household income and costs related to COM reported in the sociodemographic questionnaire, we found that up to 41% of COM patients have a household monthly income between $282 and $563.90 USD ($925,251 to $1,850,175 COP). Besides, this population spent an average of 12.08% of their household monthly income, and up to 30.19% of their income over 6 months, on COM related costs. Although low-income populations in Colombia have access to government health subsidies, these insurances do not cover all COM treatment costs [42].

Indeed, expenses related to transportation from other regions to specialized health centers, lodging, and some medicines used as part of COM management are not covered by health insurance and must be paid directly from patients' incomes. COM patients reported a median

of 5 visits to the specialist in the last 6 months (IQR: 3 to 8), thus transportation expenses in this group of patients are significant. Regardless of this large number of ENT consultations the median length of COM was 26.13 years, hence this group of patients spent a long period without access to proper surgical treatment. This circumstance could be explained by the bureaucratic difficulties with the health insurance for surgical authorizations, denoting some of the health-care system access barriers in Colombia.

This information evidences the economic burden of COM which has been previously described in high-income countries [2, 22]. Besides, it has been stated that direct payment (or out-of-pocket spending) for health services is an inefficient and regressive source of financing that implies an unstable flow of resources, and constitutes a barrier that delays healthcare access [23]. This situation is particularly important in chronic diseases such as COM, which is more frequent in the poorest populations, with low socioeconomic status, and which come from low to middle-income countries [2, 5, 12, 15, 33, 34]. For these vulnerable populations, average expenses of 12% spent on COM represent a significant fraction of their household income. Further studies regarding the burden of COM in low to middle-income countries, especially in Latin America, are needed.

Quality of life in patients with COM was assessed using the Spanish version of the COMQ-12 questionnaire [21]. Previous studies report that patients with active COM obtained higher overall COMQ-12 scores [3, 20, 24], suggesting that the quality of life could be significantly deteriorated in patients with active COM. Similarly, we found significant differences (P <0.05) in the overall scores of the scale and the domains "severity of symptoms" and "impact of lifestyle and work" between active and inactive COM. Thus, our results are consistent with previous research studies.

Among the strengths of this study, we should mention that the evaluation of the otoscopic findings was carried out by two otologists with more than 30 years of experience in this field so the otoscopic results are valid and reliable. An additional strength of our study is the sample size: the minimum sample size estimated for this study was 278 (139 patients with COM, and 139 control adults) and we achieved a sample size of 344 (200 patients with COM, and 144 control adults). Besides, the robust logistic regression analysis to determine the associated factors of COM was performed including the number of total impaired ears, equivalent to 400 affected ears. This sample size granted a significant statistical power of the tests we carried out. The Spanish version of COMQ-12 questionnaire for the evaluation of patients' quality of life is also highlighted since this scale has international recognition and previous studies suggest that it obtained reliable and homogeneous statistical results [3, 21, 24].

Finally, among the limitations of this study, we remark that this is a cross-sectional study in which an association, but no causal relationship, can be established between the variables [43]. Besides, we should mention that the audiometric tests were brought by the patients to the consult and were performed in multiple external audiometric clinics. As previously mentioned, audiometric results from different clinics may show significant variations of sound pressure between audiometers (between test variation) [41]. Thus, this should be considered as a limitation of the study. Further studies assessing associated factors, quality of life, and cost related to COM through standardized strategies are needed in low to middle-income countries. This information would be essential to support the development of preventive and therapeutic public health strategies needed to reduce COM prevalence, particularly in low to middle income countries.

## Conclusion

The associated factors for COM found in this study such as medium and high socioeconomic, female sex and URTI are consistent with previous studies. A large percentage of COM patients

spend a significant fraction of their household income on expenses related to this disease. The validated Spanish language COMQ-12 questionnaire allowed us to determine the quality of life deterioration triggered by COM symptoms. However, further studies using standardized methods for the assessment of associated factors, quality of life, and the financial burden of this disease are required.

## Acknowledgments

Special thanks to Eliana Parra and Emily Harmon for their writing assistance and technical editing. We would like to thank all the Otolaryngology Residents from Fundación Universitaria de Ciencias de la Salud for their support in collecting data.

## Author Contributions

**Conceptualization:** Daniel Peñaranda, Ana M. Otoya-Tono, Lorena Gutiérrez- Velasco, Augusto Peñaranda.

**Data curation:** Lucia C. Pérez-Herrera, Sergio Moreno-López.

**Formal analysis:** Lucia C. Pérez-Herrera, Daniel Peñaranda, Sergio Moreno-López.

**Funding acquisition:** Daniel Peñaranda, Juan Manuel García, Augusto Peñaranda.

**Investigation:** Lucia C. Pérez-Herrera, Ana M. Otoya-Tono, Lorena Gutiérrez- Velasco, Juan Manuel García, Augusto Peñaranda.

**Methodology:** Lucia C. Pérez-Herrera, Sergio Moreno-López, Augusto Peñaranda.

**Software:** Sergio Moreno-López.

**Supervision:** Ana M. Otoya-Tono, Juan Manuel García, Augusto Peñaranda.

**Validation:** Lucia C. Pérez-Herrera, Daniel Peñaranda, Sergio Moreno-López, Augusto Peñaranda.

**Visualization:** Lucia C. Pérez-Herrera.

**Writing – original draft:** Lucia C. Pérez-Herrera, Daniel Peñaranda, Sergio Moreno-López, Augusto Peñaranda.

**Writing – review & editing:** Lucia C. Pérez-Herrera, Daniel Peñaranda, Sergio Moreno-López, Ana M. Otoya-Tono, Lorena Gutiérrez- Velasco, Juan Manuel García, Augusto Peñaranda.

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
