## [Decision Letter · Decision Letter 0]

2 Nov 2020

PONE-D-20-31693

Associated factors, health-related quality of life, and reported costs of chronic otitis media in adults at two otologic referral centers in a middle-income country

PLOS ONE

Dear Dr. Peñaranda,

Thank you for submitting your manuscript to PLOS ONE. After careful consideration, we feel that it has merit but does not fully meet PLOS ONE’s publication criteria as it currently stands. Therefore, we invite you to submit a revised version of the manuscript that addresses the points raised during the review process.

We look forward to receiving your revised manuscript.

Kind regards,

Rafael da Costa Monsanto, M.D.

Academic Editor

PLOS ONE

Journal Requirements:

2. In your Methods section, please provide additional information about the participant recruitment method and the demographic details of your participants. Please ensure you have provided sufficient details to replicate the analyses such as:

a) a description of any inclusion/exclusion criteria that were applied to participant recruitment,

b) a description of how participants were recruited, and

c) descriptions of where participants were recruited and where the research took place.

Reviewers' comments:

Reviewer's Responses to Questions

**Comments to the Author**

1. Is the manuscript technically sound, and do the data support the conclusions?

Reviewer #1: Yes

Reviewer #2: Yes

2. Has the statistical analysis been performed appropriately and rigorously? 

Reviewer #1: Yes

Reviewer #2: Yes

3. Have the authors made all data underlying the findings in their manuscript fully available?

Reviewer #1: Yes

Reviewer #2: Yes

4. Is the manuscript presented in an intelligible fashion and written in standard English?

Reviewer #1: Yes

Reviewer #2: Yes

5. Review Comments to the Author

Reviewer #1: This manuscript “PONE-D-20-31693” is a study designed to address the epidemiological, clinical, sociodemographic, and economic aspects of chronic otitis media whose scientific methodology is well designed for the population analysis of Bogota. I must emphasize that the statistical analysis is well structured with sample calculation, multivariate analysis, which shows the sample power of this study.

I congratulate this initiative because we really need studies that assess the burden of chronic otitis media and its impact on quality of life that can help the development of preventive and therapeutic public health strategies needed to reduce chronic otitis media prevalence in Latin American countries.

However, we would like to point out some aspects that need clarification.

1) In the method in line 139 of the definition of chronic otitis media, it would be important to clarify that this diagnosis comprises a very broad spectrum with different clinical characteristics and that aspects of activity and inactivity were also addressed. This was only addressed in the otoscopic evaluation, leaving this definition very imprecise.

2) How the socioeconomic level of the sample was assessed as well as the level of education and this study was carried out in two university centers and there was an extremely small number of individuals in the upper-middle class as well as in the higher levels of education. This data may be a potential bias in the sample as it is not described whether these two services assess all social classes or if they preferentially attend to low socioeconomic status.

3) The sentence in Line 413 of the discussion is incomplete "that suggests a mild to moderate conductive hearing loss in patients with [37–39]." compromising the understanding for this I suggest revising it.

4) Still in line 416 of the discussion, where audiometric assessments were discussed, with no differences between cases with otitis activity and inactivity being observed, this data may present an extremely important bias since all audiometries were performed in different places where it cannot be controlled. both the acoustic isolation and the dependent effect of the examiner, for this reason their findings differ from others in the literature, mainly in relation to the non-finding of mixed and sensorineural losses in their sample, which is really large. I think that these clarifications should be made with your analysis and not only when addressing the limitations of this study.

Reviewer #2: Despite CSOM being a disease that has already been vastly discussed in medical literature, indeed there are few published articles exposing the financial burden, impact on quality of life and associated factors related to CSOM in low-income countries, especially in Latin America. Therefore, this manuscript may fill gaps in the current literature about this particular theme.

Regardless of the large number of variables assessed in this study, the authors seemed to have considered all the variables thoroughly. Furthermore, all the Materials and Methods section was carefully described – including the statistical analysis, which increases the reliability of the results.

In addition, this manuscript has some interesting points, such as delimitating the financial impact on the family’s income and pointing to the longer period of CSOM before medical treatment - which may reflect the difficulty in accessing specialized medical support, among other factors. These findings are particularly common in low-income countries, but surprisingly publications addressing the sociodemographic factors related to COM/CSOM are scarce. Other clinical features described in this manuscript were already published before (hearing loss, impact on quality of life using specific questionnaires, risk factors associated with CSOM), but these findings corroborated well with the medical literature and allowed to have a global view of the CSOM patient.

Finally, this study has some limitations – which were honestly pointed by the authors. In my opinion, it is a very interesting manuscript as it analyses all aspects of the burden of having CSOM.

6. PLOS authors have the option to publish the peer review history of their article (what does this mean?). If published, this will include your full peer review and any attached files.

Reviewer #1: No

Reviewer #2: **Yes: **Ana Luiza Kasemodel

---

## [Author Response · Author response to Decision Letter 0]

25 Nov 2020

-- Journal Requirements:

***Answer: Thanks for this suggestion, we checked this requirements and modified file naming. 

2. In your Methods section, please provide additional information about the participant recruitment method and the demographic details of your participants. Please ensure you have provided sufficient details to replicate the analyses such as:

a) a description of any inclusion/exclusion criteria that were applied to participant recruitment

***Answer: Lines 155-158: “subjects who met the following inclusion criteria: Two hundred patients aged 18 years-old or above with COM diagnosis who visited Hospital Universitario Fundación Santa Fe de Bogotá and Hospital de San José, agreed to participate and signed the informed consent of the study.”

Lines 172-176: Exclusion criteria were as follows: Patients who had psychiatric disorders, cognitive deficits, severe diseases affecting the middle ear (Otosclerosis, tumors or congenital abnormalities of the middle), history of head injury, meningitis, previous ear surgery, severe comorbidities (e.g. cancer, HIV), or any medical condition that would limit their ability to participate were not included. Patients who refused to sign the informed consent were not included in the study either.

b) a description of how participants were recruited, and

***Answer: Lines 134-136: The study enrolled patients seen at the Division of Otorhinolaryngology from the Hospital Universitario Fundación Santa Fe de Bogotá or the Hospital de San José August between 2018 to August 2019.

c) descriptions of where participants were recruited and where the research took place.

***Answer: Lines 136-141: “Both institutions are highly complex academic hospitals providing access to all medical specialties and are referral centers for ENT and otology patients from all over the country. These institutions are both private teaching hospitals located in Bogotá, the capital city of Colombia, a low to middle-income country in Latin America. Patients from all over the country visit these institutions seeking for otology health-care.” 

***Answer: We uploaded our figure files to PACE tool to fulfil journal requirements. 

-- Reviewer #1 comments:

This manuscript “PONE-D-20-31693” is a study designed to address the epidemiological, clinical, sociodemographic, and economic aspects of chronic otitis media whose scientific methodology is well designed for the population analysis of Bogota. I must emphasize that the statistical analysis is well structured with sample calculation, multivariate analysis, which shows the sample power of this study. I congratulate this initiative because we really need studies that assess the burden of chronic otitis media and its impact on quality of life that can help the development of preventive and therapeutic public health strategies needed to reduce chronic otitis media prevalence in Latin American countries. However, we would like to point out some aspects that need clarification.

***Answer: Thanks for highlighting the importance of our work, we really appreciate your feedback and your contributions to our manuscript. 

1) In the method in line 139 of the definition of chronic otitis media, it would be important to clarify that this diagnosis comprises a very broad spectrum with different clinical characteristics and that aspects of activity and inactivity were also addressed. This was only addressed in the otoscopic evaluation, leaving this definition very imprecise.

***Answer: Thanks for your comment, we added more information regarding how COM diagnosis was established to this section >>> Lines 144-147: “Moreover, COM diagnosis included a broad spectrum of clinical features such as hearing impairment symptoms (e.g. otalgia, tinnitus, persistent blockage of fullness of the ear, hearing loss, chronic ear drainage, balance problems), otoscopic findings to address disease activity, and pure-tone audiometric findings”

2) How the socioeconomic level of the sample was assessed as well as the level of education and this study was carried out in two university centers and there was an extremely small number of individuals in the upper-middle class as well as in the higher levels of education. This data may be a potential bias in the sample as it is not described whether these two services assess all social classes or if they preferentially attend to low socioeconomic status.

***Answer: We agree with your suggestion. We added the following information >>>> Line 192-193: “Both institutions included in this study attend population affiliated to private Health Promoting Entities which provide health insurance packages to all socioeconomic-status populations.”

3) The sentence in Line 413 of the discussion is incomplete "that suggests a mild to moderate conductive hearing loss in patients with [37–39]." compromising the understanding for this I suggest revising it.

***Answer: We checked this line and we added the missing information >>> Line 424: “…mild to moderate conductive hearing loss in patients with COM”

4) Still in line 416 of the discussion, where audiometric assessments were discussed, with no differences between cases with otitis activity and inactivity being observed, this data may present an extremely important bias since all audiometries were performed in different places where it cannot be controlled. both the acoustic isolation and the dependent effect of the examiner, for this reason their findings differ from others in the literature, mainly in relation to the non-finding of mixed and sensorineural losses in their sample, which is really large. I think that these clarifications should be made with your analysis and not only when addressing the limitations of this study.

***Answer: We agree with this suggestion and we added this information to the paragraph related to audiometric assessment analysis >>> Lines 431-435.

-- Reviewer #2 comments:

Thanks for highlighting the importance of this work, we really appreciate your feedback. We agree with all your comments regarding the scarce literature in low-income countries about COM.

---

## [Decision Letter · Decision Letter 1]

17 Dec 2020

Associated factors, health-related quality of life, and reported costs of chronic otitis media in adults at two otologic referral centers in a middle-income country

PONE-D-20-31693R1

Dear Dr. Peñaranda,

We’re pleased to inform you that your manuscript has been judged scientifically suitable for publication and will be formally accepted for publication once it meets all outstanding technical requirements.

Kind regards,

Rafael da Costa Monsanto, M.D.

Academic Editor

PLOS ONE

Additional Editor Comments (optional):

Dear authors,

Thank you for addressing all comments made by the reviewers. Please just add the final consideration made by Reviewer #1 to the final version of the manuscript.

Reviewers' comments:

Reviewer's Responses to Questions

**Comments to the Author**

1. If the authors have adequately addressed your comments raised in a previous round of review and you feel that this manuscript is now acceptable for publication, you may indicate that here to bypass the “Comments to the Author” section, enter your conflict of interest statement in the “Confidential to Editor” section, and submit your "Accept" recommendation.

Reviewer #1: All comments have been addressed

Reviewer #2: All comments have been addressed

2. Is the manuscript technically sound, and do the data support the conclusions?

Reviewer #1: Yes

Reviewer #2: Yes

3. Has the statistical analysis been performed appropriately and rigorously? 

Reviewer #1: Yes

Reviewer #2: Yes

4. Have the authors made all data underlying the findings in their manuscript fully available?

Reviewer #1: Yes

Reviewer #2: Yes

5. Is the manuscript presented in an intelligible fashion and written in standard English?

Reviewer #1: Yes

Reviewer #2: Yes

6. Review Comments to the Author

Reviewer #1: I have previously reviewed the original manuscript and thank you for thoroughly addressing the comments. The authors have provided corresponding information and the manuscript has improved overall.

I just suggest that you review line 177 where "or congenital abnormalities of the middle" is described and the term “middle” has become vague

Reviewer #2: I believe that the authors have addressed all the suggestions and questions raised by the reviewers.

I have no further considerations.

7. PLOS authors have the option to publish the peer review history of their article (what does this mean?). If published, this will include your full peer review and any attached files.

Reviewer #1: No

Reviewer #2: No

---

## [Editor Report · Acceptance letter]

23 Dec 2020

PONE-D-20-31693R1 

Associated factors, health-related quality of life, and reported costs of chronic otitis media in adults at two otologic referral centers in a middle-income country 

Dear Dr. Peñaranda:

I'm pleased to inform you that your manuscript has been deemed suitable for publication in PLOS ONE. Congratulations! Your manuscript is now with our production department. 

Kind regards, 

on behalf of

Dr. Rafael da Costa Monsanto 

Academic Editor

PLOS ONE